# Do Older People with Diabetes Meet the Recommended Weekly Physical Activity Targets? An Analysis of Objective Physical Activity Data

**DOI:** 10.3390/ijerph16142489

**Published:** 2019-07-12

**Authors:** Damiano Pizzol, Lee Smith, Ai Koyanagi, Brendon Stubbs, Igor Grabovac, Sarah E. Jackson, Nicola Veronese

**Affiliations:** 1Italian Agency for Development Cooperation, Jerusalem 9135400, Israel; 2The Cambridge Centre for Sport & Exercise Sciences, Anglia Ruskin University, Cambridge CB1 1PT, UK; 3Parc Sanitari Sant Joan de Déu, Universitat de Barcelona, Fundació Sant Joan de Déu, Sant Boi de Llobregat, 08005 Barcelona, Spain; 4Physiotherapy Department, South London and Maudsley NHS Foundation Trust, Denmark Hill, London SE5 8AZ, UK; 5Health Service and Population Research Department, Institute of Psychiatry, Psychology and Neuroscience King’s College London, De Crespigny Park, London SE5 8AF, UK; 6Department of Social and Preventive Medicine, Center for Public Health, Medical University of Vienna, 1300 Vienna, Austria; 7Department of Behavioural Science and Health, University College London, London WC1E6BT, UK; 8National Research Council, Neuroscience Institute, Aging Branch, Via Giustiniani, 2, 35,128 Padova, Italy

**Keywords:** diabetes, physical activity, physical exercise, accelerometer, depression

## Abstract

Appropriate management of diabetes mellitus (DM) includes following a healthy lifestyle, in which reaching physical activity (PA) recommendations is an important factor. Despite this, it remains unclear whether people with DM meet the recommended PA targets. We therefore aimed to investigate the proportion of older adults with DM (type 1 and 2) engaging in the recommended amount of PA per week in a cross-sectional study. PA levels were objectively measured using the GT1M ActiGraph accelerometer for seven consecutive days, and the cut-off of 150 min of moderate-to-vigorous PA (MVPA) was used. To assess the relationship between not meeting the recommendation for, and the significant factors associated with PA level (MVPA < 150 min/week), a multivariable logistic regression analysis was applied. 197 diabetic participants (mean age = 66.8 years; 46.7% males) spent only 74.5 ± 94.4 min/weekly in MVPA, and only 39 (=19.8%) reached the cut-off for sufficient PA levels. Significant correlates of not meeting the recommendation for PA levels were female sex, depressive symptoms, and age. In conclusion, only one-fifth of diabetic people reached the recommended amount of PA, suggesting that more intervention is needed to increase PA levels in this population.

## 1. Introduction

Diabetes mellitus (DM) represents a public health issue worldwide, and the World Health Organization (WHO) estimated that globally 422 million adults were living with DM in 2014, compared to 108 million in 1980 [1]. A balanced diet and regular participation in physical activity (PA) are considered of paramount importance in the prevention of DM, as these behaviors aid in the prevention of obesity and insulin resistance [2]. For example, recent evidence suggests that a balanced diet plus PA participation reduces or delays the risk of type 2 DM in people with impaired glucose tolerance [3]. Again, some authors suggest that, although all types of PA are beneficial, there is an inverse dose–response relation between PA and the risk of type 2 DM [4].

Moreover, PA is recommended as a key strategy, not only in order to prevent, but also to manage DM [1]. Specifically, higher PA levels are associated with a better metabolic profile and lowered cardiovascular diseases risk in those with DM [5]. Indeed, the American Diabetes Association recommends moderate-to-vigorous aerobic exercise for 150 min/week [6]; other authors suggest alternating low and high-intensity exercise in order to achieve a better glycemic control and to improve endothelial function [7]. Interestingly, studies comparing different PA models, both in terms of intensity [8] and duration [9], conclude that all types, although in different ways, are effective in terms of glycemic control, insulin sensitivity, body composition, blood pressure, muscular strength, and aerobic capacity.

Despite this positive relationship between PA and DM, it remains unclear the percentage of people with DM that meet the recommended PA targets. In people with DM, it is recommended to engage in at least 150 min of moderate-to-vigorous PA (MVPA) per week [6] for preventing common complications of this condition, such as cardiovascular conditions or mortality. However, these estimates are based on self-reported information regarding PA that are often biased, particularly in older people, whilst objective measures are more precise in this sense [10].

We therefore aimed to investigate the portion of people with DM engaging in the recommended 150 min of MVPA per week (measured with an accelerometer) in a large cohort of North American people with or at high risk of knee osteoarthritis. 

## 2. Materials and Methods

### 2.1. Data Source and Participants

Data were obtained from the Osteoarthritis Initiative (OAI) database. The OAI dataset included 1927 participants at wave 6 with complete data regarding accelerometry. Of this sample, 197 (=10.2%) reported a diagnosis of DM and were included in the present analyses. More than one third of the participants suffered from knee osteoarthritis. The 197 participants were recruited across four clinical sites in the United States of America (Baltimore, MD; Pittsburgh, PA; Pawtucket, RI; and Columbus, OH) between February 2004 and May 2006. Participants were included if they (1) had knee osteoarthritis (OA) with knee pain for a 30-day period in the past 12 months or (2) were at high risk of developing knee OA (e.g., they were overweight/obese (body mass index, BMI ≥ 25 kg/m^2^) or had a family history of knee OA) [11]. The data in this study were collected at wave 6, the only wave in which data regarding accelerometer were available. All participants provided written informed consent. The OAI study was given full ethics approval by the institutional review board of the OAI Coordinating Center at the University of California in San Francisco.

### 2.2. Physical Activity Levels

PA level was objectively measured using a GT1M ActiGraph accelerometer for seven consecutive days at wave 6 of the OAI. A benchmark set of cut-points reported by Troiano, [12] and previously applied to the general adult population from the National Health and Nutrition Examination Study, was used. According to this classification [12], light PA was defined as 100–2019 activity count/minute, moderate as 2020–5998, and vigorous was more than 5999. We dichotomized PA data to distinguish between participants who were vs. were not achieving the 150 weekly minutes of MVPA recommended by the American Diabetes Association (ADA) for people with DM [6].

### 2.3. Diabetes

DM was self-reported, with participants asked whether they had suffered from this condition at any point during their life. For those who reported having DM, data regarding treatment were also recorded and categorized as dietary intervention, insulin, or other anti-diabetic medications. 

### 2.4. Covariates

Other factors included were: ethnicity (white vs, other); education (college or higher vs, other); body mass index (BMI) (as a continuous measure); depressive symptoms assessed using the Center for Epidemiologic Studies Depression Scale (CES-D) [13]; smoking status (never vs. current/former); and self-reported presence of heart failure, heart attack, stroke, and cancer. Knee osteoarthritis was ascertained through radiological and clinical evaluations [14].

### 2.5. Statistical Analyses

Data on continuous variables were normally distributed, according to the Kolmogorov-Smirnov test. Data were presented as means and standard deviation (SD) values for quantitative measures and percentages for all categorical variables. 

Descriptive statistics were used to describe the proportion of people with DM in the sample who had a low level of PA. Multivariable logistic regression was used to assess the relationship between low PA level (MVPA < 150 min/week) and significant factors associated with this condition. We also applied a backward logistic regression analysis in order to find the best set of factors associated with not meeting the recommendation for PA. Multi-collinearity among covariates was assessed through variance inflation factor (VIF) [15], taking a cut-off of 2 as the criterion for exclusion. However, no covariates were excluded using this criterion. Adjusted odds (ORs) and 95% confidence intervals (CI) were calculated to estimate the strength of the associations between clinical factors and not meeting the recommendation for PA in diabetic patients. 

A *p* < 0.05 was deemed statistically significant. Analyses were performed using STATA^®^ software version 14.1 (Stata Corp LP, College station, TX, USA).

## 3. Results

Table 1 summarizes the sample characteristics. Three quarters of the sample used diet and/or oral medications for treating DM, whilst 15.2% used insulin injections. The mean age was 66.8 ± 9.0 years, and 46.7% were males. The mean BMI was in the obese range (30.7 kg/m^2^), whilst the prevalence of common medical conditions potentially affecting PA (such as cancer and cardiovascular diseases) was less than 10%. Participants with DM spent 74.5 ± 94.4 min/weekly in moderate-to-vigorous PA. Just one in five (19.8%) met the recommended level of PA suggested by the ADA for people with DM. [6].

Table 2 reports significant correlates of not meeting the recommendation for PA in diabetic patients showing that female sex (OR = 2.694; 95% CI: 1.201–6.047; *p* = 0.016), increase at one point in CESD (OR = 1.120; 95% CI: 1.039–1.206; *p* = 0.003), and age (increase in one year) (OR = 1.117; 95% CI: 1.063–1.173; *p* < 0.0001) are significantly associated with not meeting the recommendation for PA in diabetic patients. Knee OA was not a significant predictor of low physical activity in this cohort in the multivariable analysis (OR = 0.96; 95% CI: 0.54–1.48; *p* = 0.84).

## 4. Discussion

In a cohort of North American older adults with or at high risk of knee OA, just one in five people with DM was found to be engaging in the recommended 150 weekly minutes of MVPA suggested by the ADA [6]. Moreover, among significant correlates of not meeting the recommendation for PA in diabetic patients (other than female sex and age), depression seems to be relevant. 

Among all social factors, education for the management of diabetes seems to be the most important. In fact, DM self-management, including PA, is considered the single most important determinant for DM control and, in turn, is mostly influenced by literacy education [16]. Indeed, education is crucial in order to understand all aspects of DM, its complications, and the benefits from adherence to lifestyle recommendations [17]. Interestingly, in our sample, just 18.8% had a graduate degree, i.e., college or higher grade of education. Unfortunately, we have no information regarding the health education provided them from health workers, but this low graduated rate can partially explain why our research shows that a small percentage can reach the cut-off suggested by the ADA, indicating that clinicians should encourage PA in diabetic people. Nevertheless, our data suggests that it is crucial to find attractive and tailored methods to stimulate DM patients to exercise.

Another interesting finding of our study, in line with previous scientific literature, is the higher association with depressive symptoms [18]. In fact, people with a depressive disorder are known to have higher odds of not meeting the recommended PA levels, compared to people without depression [18]. Contrariwise, structured PA is demonstrated to reduce depressive symptoms [19].

To promote and improve the adherence and quality of PA in diabetic people, several approaches and strategies are proposed. The UK National Institute for Health Research suggested as an effective model exercise referral scheme interventions that consist of referring patients to third party services that prescribe and monitor tailored exercise programs [20]. Another interesting alternative approach, suggested by Yom-Tov and colleagues, is based on a computerized mobile app [21]. Although this approach requires careful integration of hardware, software, and human guidance; measurements may be less accurate than those of dedicated devices. Preliminary data suggest that continuous monitoring and personalized guidance generated by a computer can have a significant impact on patient behavior [21]. In the same direction, new studies will investigate the effectiveness of a novel smartphone-based, game-like software to promote regular, daily PA among DM patients [22]. While and if it will be found, the best way to promote PA among DM people will be crucial to improve and increase the communication between the patient and health care providers in order to have a significant impact on treatment outcomes and public health.

The study findings should be interpreted within its limitations. First, the people in wave 6 of the OAI represent only a small part of the people initially included and, of them, only 197 had DM. Thus, a selection bias is possible. Second, the OAI includes only people with knee OA, or at high risk for this condition (such as overweight or obesity), a common risk factors for not reaching recommended PA levels. Finally, the diagnosis of DM was made only by self-reported information that seems to have a poor accuracy compared to other tools, such as HbA1c or blood glucose [23].

## 5. Conclusions

In conclusion, these results suggest that, in this cohort of people affected by knee OA or at high risk for this condition, only one in five people with DM meet the recommended amount of PA, suggesting more focus is needed on this issue. Depression can be a potential target for future interventions. Future research is required to explore barriers and facilitators to PA and the best methods to encourage people with DM to become less sedentary and more active. 

## Figures and Tables

**Table 1 ijerph-16-02489-t001:** Characteristics of the participants with diabetes at wave 6 of the Osteoarthritis Initiative (*n* = 197).

Characteristics	Mean (SD)/%	Range
Diabetes treatment		
Not treated (%)	3	-
Diet (%)	75.6	-
Oral anti-diabetic medications (%)	75.6	-
Insulin (%)	15.2	-
General characteristics		
Age (years)	66.8 (9.0)	49–83
Males (%)	46.7	-
Caucasians (%)	67.5	-
Smoking (previous/current) (%)	47.2	-
College or higher (%)	18.3	-
Medical conditions		
BMI (Kg/m^2^)	30.7 (4.8)	20–47
CES-D (points)	7.7 (8.4)	0–45
Knee osteoarthritis (%)	36.6	-
Heart attack (%)	6.6	-
Heart failure (%)	6.1	-
Stroke (%)	6.1	-
Cancer (%)	7.6	-
Physical activity parameters		
Moderate-to-vigorous (minutes/weekly)	74.5 (94.4)	0–423
Meeting recommendation (%)	19.8	

**Table 2 ijerph-16-02489-t002:** Significant correlates of low physical activity level in diabetic participants.

Outcome	OR	95% CI Lower	95% CI Higher	*p*-Value
Female sex	2.694	1.201	6.047	0.016
CESD (points)	1.12	1.039	1.206	0.003
Age (years)	1.117	1.063	1.173	<0.0001

Abbreviations: CES-D: Center for Epidemiologic Studies Depression Scale; OR: odds ratio; CI: confidence interval.

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
