# Peer review of "Do Older People with Diabetes Meet the Recommended Weekly Physical Activity Targets? An Analysis of Objective Physical Activity Data"

_ijerph, 2019, doi:10.3390/ijerph16142489_

Round 1
Reviewer 1 Report
Thank you for the corrections.
Author Response
Reviewer 1
- Thank you for the corrections.
Answer: thank you for your approval
Reviewer 2 Report
The adjustment of knee osteoarthritis in the analysis of Multivariate regression is not clear!
All associations analysis (OR) must be adjusted by knee osteoarthritis (presence or absence). The authors described that these analysis were added. But, no informations about adjustment were provided (Please, What is page? line?)
In addition, is knee osteoarthritis a confounder factor of association (Odds ratio) between diabetes and physical activity?
Author Response
Reviewer 2
The adjustment of knee osteoarthritis in the analysis of Multivariate regression is not clear!
All associations analysis (OR) must be adjusted by knee osteoarthritis (presence or absence). The authors described that these analysis were added. But, no informations about adjustment were provided (Please, What is page? line?)
In addition, is knee osteoarthritis a confounder factor of association (Odds ratio) between diabetes and physical activity?
Thank you for your comment. Knee OA was initially put as covariate, but using a backward log regr analysis, it was not a significant predictor of low PA. We better explained (above table 2) as follow: OA was not a significant predictor of low physical activity in this cohort in the multivariable analysis (OR=0.96; 95%CI: 0.54-1.48; p=0.84).
Reviewer 3 Report
The current paper aims to provide an objective assessment of physical activity data on older adults with diabetes. Furthermore, the authors aim to assess the proportion of the sample that achieves physical activity recommendations. The authors should be commended for their efforts on providing empirical evidence on an important topic.
Abstract:
· Lines 30-32: The sentence is challenging to take away the main message. Are the significant factors being evaluated significant factors associated with not meeting the PA recommendations or are the significant factors being evaluated significant factors associated with DM?
· Line 32: Greatly clarity could be provided by specifying “this condition” as DM or “this condition” as “not meeting PA recommendations”
Introduction:
· Line 58: greater clarity can be provided to the reader by specifying what “this” refers to “health-related benefits of physical activity”.
· Line 58: “if people” you are really interested in the proportion of the sample meets the PA recommendations (not if or if not)
Methods:
· Line 91: Unclear if “college or higher” requires an earned college degree or completed some college.
Results:
· Line 114: the word “on” should be replaced with “from”
o Also, what percentage of the sample was “at high risk of developing knee OA” or “had a family history of knee OA”
· Because the paper relates to providing an objective assessment of physical activity behavior in older diabetics and the sample is being drawn from individuals with or at risk of knee osteoarthritis, the paper could be strengthened by including data on the proportion of individuals meeting and not meeting PA recommendations by having OA or being at risk of OA, depending on the number of individuals in each category.
Discussion:
· Line 138: Was education included in the multivariate logistic regression analysis? No data are presented in the results section to support this statement.
· Line 168: “optimal” should likely be replaced with “recommended”
Author Response
Reviewer 3
The current paper aims to provide an objective assessment of physical activity data on older adults with diabetes. Furthermore, the authors aim to assess the proportion of the sample that achieves physical activity recommendations. The authors should be commended for their efforts on providing empirical evidence on an important topic.
Abstract:
- Lines 30-32: The sentence is challenging to take away the main message. Are the significant factors being evaluated significant factors associated with not meeting the PA recommendations or are the significant factors being evaluated significant factors associated with DM?
- Line 32: Greatly clarity could be provided by specifying “this condition” as DM or “this condition” as “not meeting PA recommendations”
Thank you, as suggested we changed and better explained the sentence.
Introduction:
Line 58: greater clarity can be provided to the reader by specifying what “this” refers to “health-related benefits of physical activity”.
As suggested we specified the concept.
- Line 58: “if people” you are really interested in the proportion of the sample meets the PA recommendations (not if or if not)
Corrected, as you suggested.
Methods:
- Line 91: Unclear if “college or higher” requires an earned college degree or completed some college.
The people in this category have earned at least some college.
Results:
Line 114: the word “on” should be replaced with “from”
Thank you, as suggested we provided the change.
- Also, what percentage of the sample was “at high risk of developing knee OA” or “had a family history of knee OA”
People in these categories are those without knee OA, i.e. 64.4% of the whole sample.
- Because the paper relates to providing an objective assessment of physical activity behavior in older diabetics and the sample is being drawn from individuals with or at risk of knee osteoarthritis, the paper could be strengthened by including data on the proportion of individuals meeting and not meeting PA recommendations by having OA or being at risk of OA, depending on the number of individuals in each category.
We kindly disagree with this request. The aim of our work is to determine the level of physical activity in people having diabetes. Knee OA, in our sample was not significantly associated with low physical activity, as reported in Table 2 and in the Results section.
Discussion:
- Line 138: Was education included in the multivariate logistic regression analysis? No data are presented in the results section to support this statement.
Sorry for this misunderstanding. In this case we did not mean education in the sense of school earned, but the education regarding diabetes management. We have better explained this point.
- Line 168: “optimal” should likely be replaced with “recommended”
Done.
This manuscript is a resubmission of an earlier submission. The following is a list of the peer review reports and author responses from that submission.
Round 1
Reviewer 1 Report
This is a very good straightforward study, I only have a few comments/questions.
Why did you want the participants to have knee pain?
In table 1 “Not treated”
Line 141 seems strange. Do you have data from this study that supports this claim, since this was a covariate?
Line 145 “had a graduated degree”? please correct. Does this mean graduate degree as in Masters or higher?
Reviewer 2 Report
-Whats is the sample size?
-Whey only participants were included if they had knee osteoarthritis? In fact, the authors did not aim to investigate the proportion of people with diabetes, but with knee osteoarthritis. Therefore, the conclusion (one-fifth of diabetic people reached the recommended amount physical activity level), not meet a truly data.
-Multivariable logistic regression must be adjusted by knee osteoarthritis (the major variable associated to physical activity and diabetes).
-The table 2 is wrong, once described the depression scale, female and age as associated factor to diabetes. But, not presence or absence of diabetes. All variables (depression scale, sex and age), must be confounders variables and therefore, used as adjust variable.
Reviewer 3 Report
Comments and suggestions for authors:
The current papers seeks to provide objective physical activity data on older adults with diabetes. Additionally, the investigators aim to assess compliance with physical activity guidelines in older adults with diabetes. The authors should be commended for their efforts.
• Diabetes mellitus is abbreviated DM in line 41 but the abbreviation is not used consistently throughout the paper, bouncing back between DM and diabetes. The same comment can apply to physical activity and PA and moderate-to-vigorous physical activity and MVPA. Once the abbreviation is established, the abbreviation should be used throughout.
Abstract:
• The introduction and purpose statement should specify the current study is focused on older adults.
• No differentiation in the type of diabetes mellitus is included. Does the current paper include both Type I and Type II diabetics?
Introduction:
• Line 51—52: The reference [6] provided does not align with the ADA Position Stand Paper [10]
• Line 60—61: It may be worth noting mortality, in addition to cardiovascular complications.
• Lines 61—63: References for the statements should be included. Additionally, the self-reported physical activity information could be included.
• Line 65—66: The title of the paper and the introduction relate to diabetes, yet, the purpose statement indicates the paper relates to individuals with or at risk of knee osteoarthritis.
Methods:
• Physical Activity Assessment: Was accelerometer wear time compliance evaluated? If so, what were the compliance standards? This should be included.
• Analyses should account for knee osteoarthritis
Results:
• Line 117: wording issues “whilst the 15.2% insulin injections.”
• Line 118: No SD for BMI
• Table 1: Identifier of “Graduate Degree” is different than verbiage in methods section.
• Presenting data on time in moderate-intensity, vigorous-intensity, and moderate-to-vigorous intensity physical activity would be of interest to the reader and should be included.
Discussion:
• Line 136: The sentence indicates PA in individuals with type 2 DM is less than PA in individuals without type 2 DM, based on data in the current study. This was not the purpose of the study and was not a comparison made in the current study.
• The current study included an education variable but no data related to comparing physical activity by educational level is included. Thus, Line 141 and the subsequent sentences seem a bit displaced in the context of the current study.
• Line 145: ‘graduated degree’ terminology is different than terminology used in methods.
• Overall, the content and flow of the discussion is a bit challenging to follow. For example, the second paragraph is intended to focus on the relationship between education and physical activity levels, but the authors seem to unnecessarily focus on health education rather than just overall education levels. Additionally, in the last sentence of the third paragraph, the authors include a statement about structured physical activity but literature comparing structured vs. unstructured is not included elsewhere in the paper.
• There are several grammatical errors that need to be resolved.